# Performance Enhancement of All-Inorganic Carbon-Based CsPbIBr_2_ Perovskite Solar Cells Using a Moth-Eye Anti-Reflector

**DOI:** 10.3390/nano11102726

**Published:** 2021-10-15

**Authors:** Wensheng Lan, Dazheng Chen, Qirui Guo, Baichuan Tian, Xiaoping Xie, Yibing He, Wenming Chai, Gang Liu, Peng Dong, He Xi, Weidong Zhu, Chunfu Zhang

**Affiliations:** 1State Key Discipline Laboratory of Wide Band Gap Semiconductor Technology, Xidian University, Xi’an 710071, China; lws85626637@163.com (W.L.); guoqirui1014@163.com (Q.G.); tbcxd0515@yeah.net (B.T.); heyibing0507@163.com (Y.H.); 15009205592@163.com (W.C.); hxi@xidian.edu.cn (H.X.); wdzhu@xidian.edu.cn (W.Z.); 2Qinghai Huanghe Hydropower Development Co., Ltd., Xining 810008, China; xiexiaoping@spic.com.cn (X.X.); liugang@spic.com.cn (G.L.); dongpeng@spic.com.cn (P.D.)

**Keywords:** inorganic CsPbIBr_2_ solar cell, moth-eye anti-reflector, OrmoStamp, nano-imprinting, FDTD

## Abstract

All-inorganic carbon-based CsPbIBr_2_ perovskite solar cells (PSCs) have attracted increasing interest due to the low cost and the balance between bandgap and stability. However, the relatively narrow light absorption range (300 to 600 nm) limited the further improvement of short-circuit current density (J_SC_) and power conversion efficiency (PCE) of PSCs. Considering the inevitable reflectance loss (~10%) at air/glass interface, we prepared the moth-eye anti-reflector by ultraviolet nanoimprint technology and achieved an average reflectance as low as 5.15%. By attaching the anti-reflector on the glass side of PSCs, the J_SC_ was promoted by 9.4% from 10.89 mA/cm^2^ to 11.91 mA/cm^2^, which is the highest among PSCs with a structure of glass/FTO/c-TiO_2_/CsPbIBr_2_/Carbon, and the PCE was enhanced by 9.9% from 9.17% to 10.08%. The results demonstrated that the larger J_SC_ induced by the optical reflectance modulation of moth-eye anti-reflector was responsible for the improved PCE. Simultaneously, this moth-eye anti-reflector can withstand a high temperature up to 200 °C, and perform efficiently at a wide range of incident angles from 40° to 90° and under various light intensities. This work is helpful to further improve the performance of CsPbIBr_2_ PSCs by optical modulation and boost the possible application of wide-range-wavelength anti-reflector in single and multi-junction solar cells.

## 1. Introduction

Since the first successful preparation by Miyasaka et al. [1], organic-inorganic hybrid perovskite solar cells (PSCs) have achieved an explosive increase of power conversion efficiency (PCE) from 3.8 to 25.5% during ten years [2,3,4,5,6,7], closing to the PCE of crystal silicon solar cells, and have been recognized as the next-generation cost-effective photovoltaic technology. However, the relatively poor thermos stability and humidity stability of PSCs with organic ions (e.g., MA^+^ and FA^+^) impeded their fast commercialization process [8,9]. An effective solution is to replace the organic ions by inorganic ions (e.g., Cs^+^ and Rb^+^), such as CsPbI_3_, CsPbI_2_Br, CsPbIBr_2_, and CsPbBr_3_ [10,11,12,13,14], which were called as all-inorganic PSCs and have attracted more and more research interest. Although CsPbI_3_ and CsPbI_2_Br show a relatively narrow bandgap of about 1.73 and 1.90 eV, the phase instability at room temperature is unfavorable for the long-term operation of PSCs [15,16,17]. CsPbBr_3_ achieved the best environment stability; however, the large bandgap (~2.3 eV) limited its light absorption edge to less than 540 nm [18,19]. While the all-inorganic CsPbIBr_2_ PSCs not only possess excellent stability under ambient air atmosphere with the controlled relative humidity (RH) of ~40% and the temperature of ~25 °C, which can keep 94% of its initial PCE after 35 days and 91% of the value after longer test of 60 days [20], but also a bandgap of 2.05 eV and a theoretical efficiency as high as 22.1% [21,22,23,24]. Thus, the balance of bandgap and stability made CsPbIBr_2_ a better choice to fabricate all-inorganic PSCs.

Due to the facile and low-cost solution deposited method, carbon-based CsPbIBr_2_ PSCs have drawn great attention since the first report in 2017 by Liang et al. [20]. The carbon paste electrodes, composed of carbon black and graphite flakes, have the advantages of cheap, chemical stability, water-resistance, insensitivity to ion migration, Schottky contact at the perovskite/carbon interface for the hole-transport layer (HTM)-free devices [25,26], and the geometry of glass/FTO/c-TiO_2_/CsPbIBr_2_/Carbon has been chosen as the most typical device structure. To enhance the performance of carbon-based CsPbIBr_2_ PSCs, we have proposed the intermolecular exchange strategy (CsI treatment) [27], flux-mediated growth strategy (CH_3_NH_3_Br incorporation) [20], light-processing technology [28], and aged precursor method [27] to grow full-coverage, large-grained, pure-phase CsPbIBr_2_ films, which boosted the open circuit voltage and record PCE of devices to 1.323 V and 10.82%. Besides the CsPbIBr_2_ film itself, the interface modifications such as CsBr-modified TiO_2_ electron transport layer (ETL) [15], nanosized diamond modification [29], c-TiO_2_/m-TiO_2_ composite layer [30], NiO/ALD-TiO_2_ bilayer [31], and quinoline sulfate passivation [32] have been used to suppress the carrier recombination and the improve carrier transportation, the corresponding PSC obtained a fill factor (FF) exceeding 72% and the open-circuit voltage (V_OC_) beyond 1.3 V. However, the short circuit current density (J_SC_) of carbon-based CsPbIBr_2_ PSCs was lower than 11 mA/cm^2^ in most reports, and the external quantum efficiency (EQE) values of PCS always showed an obvious decrease from 400 to 600 nm. Although the ion doping strategy (Sn^2+^ or Ba^2+^) [25,33,34,35] has been proved effective to improve the J_SC_ of PSCs by decreasing the bandgap of perovskite, the optical loss before sunlight enters a device has been rarely discussed.

In fact, the optical loss caused by the reflection at the air/glass interface has consumed about 10% of the incident photons [36,37], which will directly reduce the most possible maximum J_SC_ by 10% for PSCs. At present, antireflection techniques mainly include monolayer or multilayer antireflection films by evaporation and periodic nanostructures by etching or nanoimprinting process [38,39,40,41]. Inspired by the eye of a moth, the bionic moth-eye periodic nanostructures could induce a gradual refractive index gradient at the surface and demonstrate a wide-range-wavelength antireflection property, which provides an optical modulation method to improve the PCE of solar cells [42,43,44,45,46,47]. In 2019, M. Choi et al. fabricated an inverted moth-eye nanostructure (300 and 1000 nm) with polydimethylsiloxane (PDMS, n ≈ 1.43) using soft lithography, and the resulted J_SC_ (PCE) of CH_3_NH_3_I_x_Br_3-x_ PSC were increased from 23.83 mA/cm^2^ (19.66%) to 25.11 mA/cm^2^ (20.93%), respectively [46]. In 2021, considering the critical effect of the front contact of PSCs on optoelectronic performance of itself, Hossain et al. came up with multilayer nanoholes and nanodomes for high-efficiency perovskite single-junction and perovskite/perovskite tandem solar cells (PVK/PVK TSCs). The result revealed that the optimized nanophotonic front contact results in PCEs >23% and >30% for single-junction PSCs and PVK/PVK PSCs, respectively [48]. It is known that the optical properties of moth-eye structures can be adjusted by the shape, size, period, and the material; also considering the relatively narrower light absorbance range, the systemic study of moth-eye-anti-reflected carbon-based CsPbIBr_2_ PSC is essential to further improve the device photovoltaic performance.

In this work, the OrmoStamp was first used to fabricate a moth-eye anti-reflector by nano-imprinting to enhance the J_SC_ and PCE of CsPbIBr_2_ PSCs by attaching it on the glass side of device [49,50,51]. The OrmoStamp was an inorganic-organic hybrid polymer for fabricating transparent stamps in nanoimprint lithography, which has advantages of excellent mechanical and flexible properties, ultra-high resolution capabilities (sub-10 nm), high transparency for UV and visible light, high chemical and physical stability, moderate volume shrinkage during cross-linking, and long stamp lifetime [52,53]. Simultaneously, the influence of temperatures, incident angles, and light intensities on the performance of CsPbIBr_2_ PSCs with moth-eye anti-reflector have been investigated to evaluate its effectiveness in practical applications. What is more, the moth-eye nanostructure with different ratios of depth to width, line widths and periods, and the size parameters were optimized by FDTD simulation. This work is helpful to further improve the performance of CsPbIBr_2_ PSCs by optical modulation and boost the possible application of wide-range-wavelength anti-reflector in perovskite/silicon tandem solar cells.

## 2. Materials and Methods

### 2.1. Materials

Ultradry lead bromide (PbBr_2_, 99.999%), ultradry cesium iodide (Csl, 99.998%), ultradry lead chloride (PbCl_2_, 99.998%), anhydrous methanol (99.9%), and DMSO (≥99.9%, ACS reagent) were purchased from Alfa-Aesar (Haverhill, MA, USA). Absolute alcohol (AR, 99.7%) and acetone (AR, 99.0%) were supplied by Sinopharm Chemical Reagent Co., Ltd., (Shanghai, China). CH_3_NH_3_Br (purity >99.8%) was received from Xi’an Polymer Light Technology Corp, and conductive carbon paste (ZF-G03-04-2) was supplied by Shanghai MaterWin New Materials Co., Ltd., China. Fluorine-doped tin oxide (FTO) substrates (TEC8, 8Ω/sq) were purchased from Pilkington TEC Glass (Hong Kong, China). OrmoStamp and 1H, 1H, 2H, 2H-full fluorooctyltrichlorosilane (FOTS, 97%) were procured from Shanghai NTI Co., Ltd., Shanghai, China. PET flexible transparent substrate and glass substrate were purchased from Liaoning Yike Precision New Technology Co. Ltd. China. Silicon template (HT-AR-02XS, Temicon GmbH, Dortmund, Germany) was supplied by German Tech Co. Ltd. (Beijing, China).

### 2.2. Preparation of CsPbIBr_2_ PSC

A typical structure of glass/FTO/c-TiO_2_/CsPbIBr_2_/Carbon was fabricated in this work. Of this, 70 μL CsPbIbr_2_ precursor, 330.0 mg PbBr_2_, 27.8 mg PbCl_2_, and 285.0 mg CsI were dissolved in 1 mL DMSO, was spin-coated on the c-TiO_2_/FTO/glass at 1500 rpm for 30 s. Then, the spin coating speed was adjusted to 5000 rpm for 120 s, and a drop of CH_3_NH_3_Br solution was dropped using a dropper within 90–100 s. Then heat annealing is carried out at 200 °C for 30 min, and the temperature is naturally cooled to room temperature. For the detailed preparation process, please refer to the paper of our research group; the resulted PSCs possesses both high PCE and good stability under thermal treatment, long-term storage, and continuous light illustration [20].

### 2.3. Fabrication of Moth-Eye on the Device

Here the OrmoStamp (micro resist technology GmbH, Dortmund, Germany, n ≈ 1.5) [49,50,51], an ultraviolet-curable acrylic liquid medium with a viscosity of 0.41 Pa·s and a dielectric constant of about 6.8 at room temperature, has been used to prepare the moth-eye nanostructure. In detail, we place the 4 × 4 cm^2^ silicon template with moth-eye nanostructure etched by electron beam in a vacuum dryer with a pressure of 10 mbar and a volume of 8 L, and add 80 uL FOTS and keep for 30 min to fully volatilize. Finally, annealed at 130 °C for 30 min to complete the hydrophobic treatment of the silicon template. Place the hydrophobically treated silicon template flat on the glass substrate, and use an injection needle to take the UV-curing glue OrmoStamp, and put 20 drops on the silicon template. The hydrophilic PET flexible transparent substrate is slowly attached from one end to the other end, and it is irradiated with 365 nm ultraviolet light for 5–10 minutes to make it completely cured. Finally, the soft stamp is peeled from the silicon template.

The glass side of the PSC was treated with O_2_ plasma under the conditions of 400 W and 400 sccm for 3 min to complete the hydrophilic treatment. Place the processed PSCs on the glass substrate in the yellow light chamber, and use the same process steps as the preparation of the soft stamp to transfer the moth-eye nanostructure on the PET soft template to the glass surface of the PSC to obtain the moth-eye nanostructure PSCs. The schematic diagram of all preparation processes can be found in Appendix A.

### 2.4. Characterization

The surface morphology of moth-eye was investigated by use of a Zeiss Supra-40 field-emission scanning electron microscope (SEM) instrument (Mexico, MO, USA). Ultraviolet-visible(UV-vis) spectra were measured on a PerkinElmer Lambda 950 spectrophotometer. The EQE spectrum was measured by employing a 150 W xenon lamp (Oriel) equipped with a monochromator (Cornerstone, Richmond, VA, America) as monochromatic light source. Current density-voltage(J-V) curves of cells were measured with a Keithley 2450 source measurement unit under standard AM 1.5 G illumination (Crowntechinc., EASISOLAR-50-3A, Indianapolis, IN, America). The active area of cells is standardized to 0.09 cm^2^. A scan rate of 50 mV/s was adopted in the J–V measurement. The transient photocurrent (TPC) and the transient photovoltage (TPV) curves were recorded by a digital oscilloscope (Tektronix, D4105, Beaverton, OR, America) on a self-built system with a modulated 532 nm pulse laser as excitation. The water contact angles (SCI3000F) were tested to character surface hydrophobicity. All the above measurements were carried out in ambient air atmosphere.

## 3. Results

The surface topography and optical properties of nano-imprinted bionic moth-eye structure was studied first. From the SEM image in Figure 1a, the nanostructure was well-defined and the statistical results showed a diameter (line width) about 200 nm and a period about 300 nm, which are consistent with the size of silicon template (line width, period, and depth are 200, 300, and 100 nm respectively). The transmission and reflection spectra of glass and moth-eye/glass samples under room temperature (RT) and vertical illumination (VI) conditions were displayed in Figure 1b,c. It can be observed that, in the wavelength range from 300 to 800 nm, the average transmittance and reflectance of bare glass are 88.01 and 9.25%, while that of moth-eye/glass sample are 91.43 and 5.15%, respectively. Compared to the bare glass, the introduction of moth-eye nanostructure realized an increased transmittance by 3.42% and reduced reflectance by 4.1%. It should be noted that the slightly lower transmittance of moth-eye/glass at the range of 300–360 nm is due to the parasitic absorption of OrmoStamp itself. Consequently, considering the main absorption range of CsPbIBr_2_ material (300–600 nm), the moth-eye nanostructure could be used as an anti-reflector to decrease the light reflection loss and enhance the light absorption of CsPbIBr_2_ PSCs.

To verify the effectiveness of moth-eye anti-reflector, they were attached on the glass side of PSCs with a typical structure of glass/FTO/c-TiO_2_/CsPbIBr_2_/Carbon (shown in Figure 1d), and the measured absorption spectra and EQE of PSCs were present in Figure 1e,f, respectively. It is clear that the optical absorption cutoff edge of CsPbIBr_2_ PSCs (without carbon electrode) was located at about 600 nm, and the absorption ability increased obviously when a moth-eye anti-reflector was used. Further from Figure 1f, the inter integrated current density (J_inte_) of the PSCs was enhanced by 9.91% from 10.39 mA/cm^2^ to 11.42 mA/cm^2^ and the significantly higher EQE values can be seen in the range from 400 to 600 nm. The slightly low EQE at the short wavelengths resulted from the relatively low transmittance (seeing Figure 1b). Therefore, the moth-eye nanostructure is favored to increase the number of photons transmitted to the active layer and improve the photon-electron conversion efficiency of PSCs. This can be further confirmed from the current density–voltage (J-V) curves in Figure 2a and photovoltaic parameters in Table 1. The reference PSCs without a moth-eye anti-reflector showed a PCE of 9.17% with the J_SC_ = 10.89 mA/cm^2^, V_OC_ = 1.28 V, and FF = 65.83%; while for the moth-eye modified device with an active area of 8.5 mm^2^, an obvious increased J_SC_ of 11.91 mA/cm^2^ and PCE of 10.10% had been achieved. The percentage improvement of J_SC_ and PCE were 9.37 and 9.92%, respectively. Consequently, the improved J_SC_ is responsible for the higher PCE, which can be further verified by the statistical results of PCE, J_SC_, V_OC_, and FF from twenty couples of PSCs shown in Figure 3. Additionally, the steady output current density at the voltage bias of the maximum power point for PSCs were shown in Figure 2c, and the device with a moth-eye anti-reflector demonstrated a better continuous light stability than that of reference device. It should be noted that, as the moth-eye anti-reflectors was placed at the surface of glass substrate where the incident light entered the devices, the moth-eye structure can also be used in the PSCs with a hole transport layer and metal electrode. In this case, compared to the carbon-based PSCs, the photons reflected by the back metal may be further absorbed by CsPbIBr_2_ layer and contribute to the improvement of photocurrent in PSCs.

To understand the physic reason of improved device performance, the photocurrent density (J_ph_)-effective voltage (V_eff_) curves, TPC, TPV, and Nyquist curves have been carried out. Here the J_ph_ is expressed as J_ph_ = J_light_ − J_dark_, in which J_light_ and J_dark_ denote current densities under illumination and dark conditions; the V_eff_ is represented as V_eff_ = V_0_ − V_a_, in which V_0_ denotes the voltage when J_ph_ is equal to 0, and V_a_ is the applied bias voltage. From Figure 2b, the J_ph_ increases linearly when the V_eff_ is less than 0.2 V and tends to saturation current density (J_sat_) at larger V_eff_ than 1 V. Particularly, the device with the moth-eye anti-reflector exhibits a remarkably higher J_light_ of 11.67 mA/cm^2^ than that of reference device (10.50 mA/cm^2^), thus the improved J_ph_ values dominated the enhancement of J_SC_ and PCE for CsPbIBr_2_ PSCs with a moth-eye anti-reflector. In addition, from Figure 2d–f, the photocurrent and photovoltage decay times, and the impedance spectra are very close for PSCs with and without anti-reflector, thus the introducing of moth-eye nanostructure has almost no influence on the carrier transport and recombination processes in the devices. Thus, the optical modulation effect arising from the moth-eye anti-reflector is responsible for the senior device performance.

Then, the influence of temperatures, incident angles, and light intensities on the performance of CsPbIBr_2_ PSCs with moth-eye anti-reflector have been investigated to evaluate the effectiveness of it in practical applications. Figure 4a showed that a temperature below 120 °C has no effect on the transmission of the moth-eye nanostructure when the angle of incidence was 90°. Above 120 °C, the transmittance started to decrease with the temperature and to be lower than that of glass at a high temperature of 240 °C. This is due to that the OrmoStamp film would crack at a high temperature (Appendix A) and the parasitic absorption of film seriously increased (Appendix A). In Figure 4b, although the reflectance gradually increased with the temperature, they were still lower than that of glass in the range from 300 to 800 nm, thus serving as an efficient anti-reflector. Figure 4c illustrated the dependence of transmittance of moth-eye/glass on the incident angles at room temperature, here the angle of incidence was defined as the angle between the incident light and the horizontal plane. It is clear that the transmittance of moth-eye/glass at the incident angle of 40° was relatively lower than the case of perpendicular incidence, but still higher than that of the bare glass. When the incident angle was 20°, the moth-eye nanostructure cannot enhance the transmittance of glass. As a result, this moth-eye anti-reflector can withstand a temperature up to 200 °C, and perform efficiently at a wide range of incident angles from 40° to 90°. Further, the corresponding evolution of photovoltaic parameters and J-V curves for CsPbIBr_2_ PSCs were displayed in Figure 5 and Appendix A, respectively. It can be observed that, under various temperature and incident angle, all the J_SC_ and PCE of devices with moth-eye anti-reflector were superior to the cases without anti-reflector, and there is no difference among the values of V_OC_ and FF. This means that within the tolerable temperatures and angles of incidence, the moth-eye anti-reflector can definitely improve the performance of PSCs by reducing the light reflection loss at air/glass interface. Additionally, due to the water contact angle of moth-eye film was higher than glass substrate (114° vs. 23° in Appendix A), the hydrophobic property of moth-eye anti-reflector attached to glass substrate would be helpful to enhance the moisture stability of CsPbIBr_2_ PSCs.

In Figure 6a–c, at different light intensity levels, the higher J_SC_ and PCE were also achieved for the device with the moth-eye anti-reflector, which indicates that the suitability of moth-eye nanostructure in CsPbIBr_2_ PSCs at various light intensity conditions. In detail, when the light intensity reduced from 100 to 10 mW/cm^2^, the values of J_SC_ and PCE decreased from 11.1 mA/cm^2^ and 8.99% to 1.9 mA/cm^2^ and 1.34%, respectively, with nearly unchanged V_OC_. It is well known that the variation of V_OC_ and J_SC_ for solar cells under different light intensity contains the information of carrier recombination. For direct recombination, the relationship between V_OC_ and light intensity will be linear in the semi-logarithmic coordinate with the slope of k_B_T/q relationship, where k_B_, T, and q are Boltzmann’s constant, Kelvin temperature, and single-electron charge respectively; the nearly identical slope (1.33 k_B_T/q vs. 1.34 k_B_T/q) in Figure 6a indicated that the introduction of moth-eye anti-reflector would not affect the carrier recombination inside the devices. On the other hand, the J_SC_ is always proportional to the light intensity (I^α^), where the α is for exponential factor and the closer to one, the smaller the degree of direct recombination [54,55,56]. As shown in Figure 6b, the PSCs with moth-eye anti-reflector presented a larger J_SC_ than reference device; and the corresponding α values were 0.878 and 0.876, which also confirms that this moth-eye nanostructure is an optical modulation to enhance the performance of CsPbIBr_2_ PSCs and have no significant influence on the carrier transport in devices. For the large-area devices (1 cm^2^), it can be seen from Figure 6d and Table 1 that the improvement of PCE for CsPbIBr_2_ PSCs with moth-eye anti-reflector resulted from the enhanced J_SC_ by 15.09%, which is in line with above discussions. In addition, we also performed the optical simulations based on finite-difference time-domain (FDTD) [39,57,58,59] to explore the more efficient moth-eye nanostructures. It is found that by adjusting the ratios of depth to width, line widths, and periods, this structure is promising to achieve better anti-reflectivity at wider wavelength range (seeing Appendix A), and thus may be used in future single and multi-junction solar cells.

## 4. Conclusions

OrmoStamp-based moth-eye anti-reflector has been prepared by nano-imprinting and successfully attached to the glass side of carbon-based inorganic CsPbIBr_2_ PSCs. The resulted J_SC_ was promoted by 9.4% from 10.89 mA/cm^2^ to 11.91 mA/cm^2^, which is the highest among PSCs with a structure of glass/FTO/c-TiO_2_/CsPbIBr_2_/Carbon, and the PCE of device was enhanced by 9.9% from 9.17 to 10.08%. It has been verified that the reduction of reflectance and increase of transmittance at air/glass interface induced by the moth-eye anti-reflector are responsible for the improvement of J_SC_ and PCE, and there is almost no influence on the V_OC_ and FF. Simultaneously, this moth-eye anti-reflector can withstand a high temperature up to 200 °C, and perform efficiently at a wide range of incident angles from 40° to 90° and under various light intensities. Additionally, the FDTD simulated composite moth-eye nanostructure showed an average reflectance below 1% in 200 to 1600 nm. This work is helpful to further improve the performance of CsPbIBr_2_ PSCs by optical modulation and boost the possible application of wide-range-wavelength anti-reflector in single and multi-junction solar cells.

## Figures and Tables

**Figure 1 nanomaterials-11-02726-f001:**
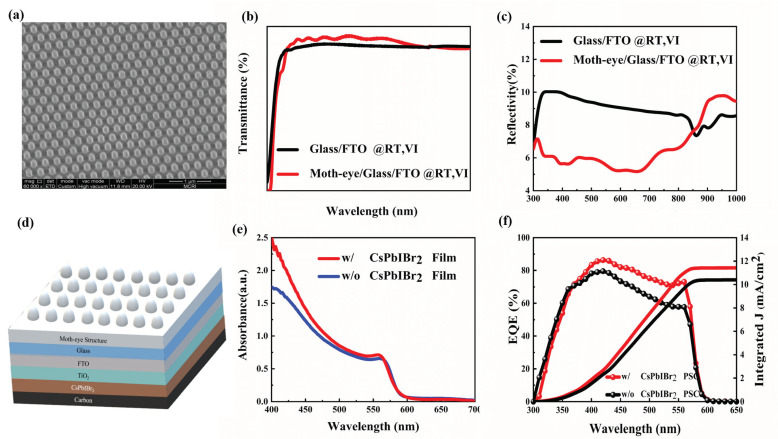
(**a**) SEM image of moth-eye nanostructures; (**b**) Transmission and (**c**) reflection spectrum of moth-eye/glass and glass under RT and VI conditions; (**d**) Schematic image of CsPbIBr_2_ PSCs with moth-eye nanostructures; (**e**) Absorption spectra of CsPbIBr_2_ film and (**f**) EQE curves and corresponding J_inte_ of CsPbIBr_2_ PSCs with and without moth-eye nanostructure.

**Figure 2 nanomaterials-11-02726-f002:**
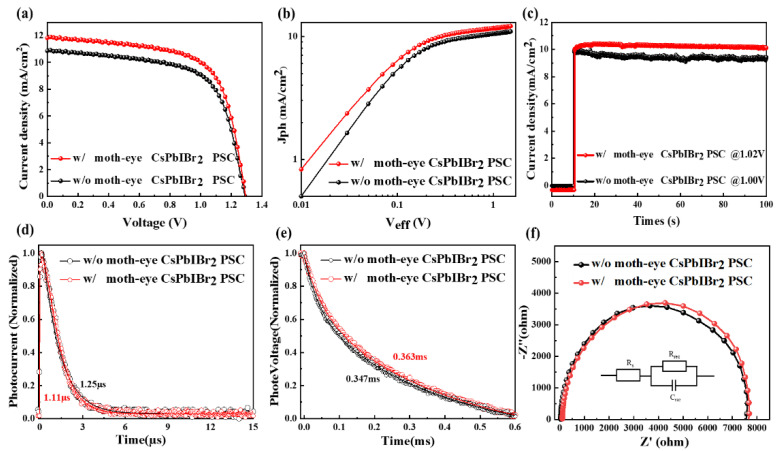
(**a**) J-V curves, (**b**) J_ph_-V_eff_ curves, (**c**) stable output characteristic curves, (**d**) TPC characteristic curves, (**e**) TPV characteristic curves, and (**f**) Nyquist curves with and without moth-eye structure device.

**Figure 3 nanomaterials-11-02726-f003:**
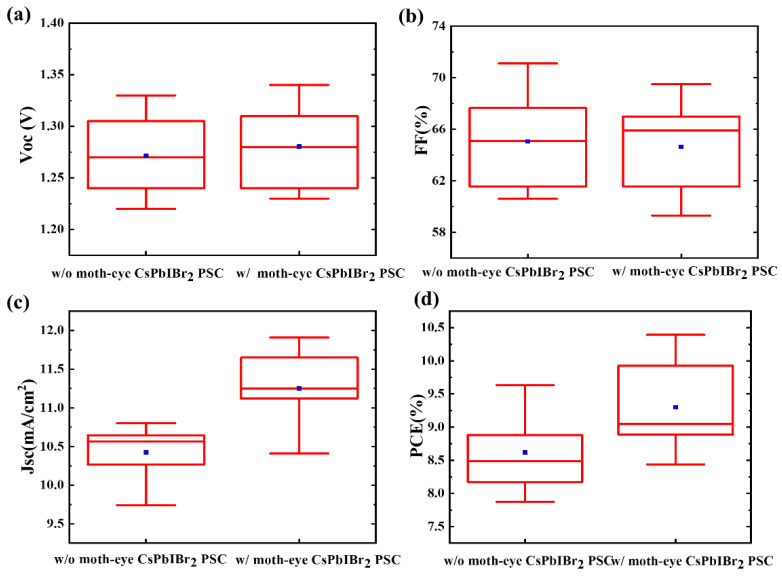
(**a**) V_OC_, (**b**) FF, (**c**) J_SC_, and (**d**) PCE statistics of 20 devices with moth-eye structure and 20 devices without moth-eye structure.

**Figure 4 nanomaterials-11-02726-f004:**
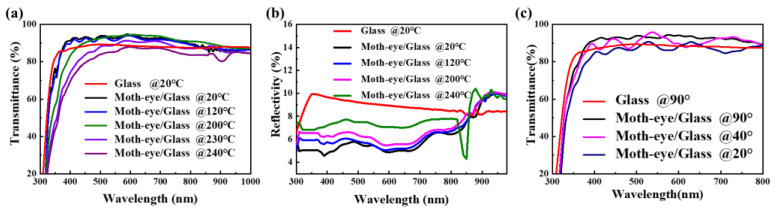
(**a**) Transmittance and (**b**) reflectance spectra of moth-eye/glass and glass at different temperatures when the light incident angle is 90°; (**c**) Transmittance spectra of moth-eye/glass and glass at incident angles of 20°, 40°, and 90°.

**Figure 5 nanomaterials-11-02726-f005:**
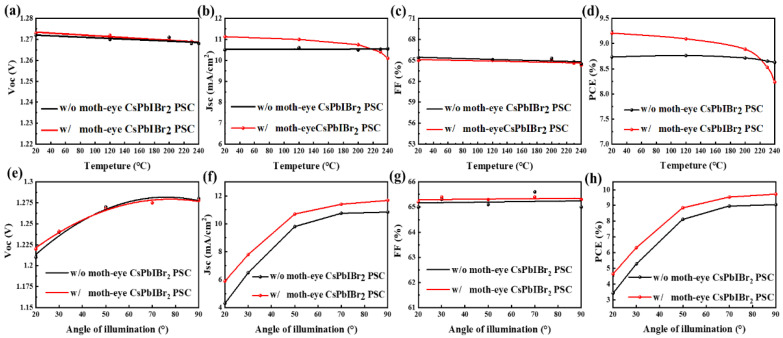
Characteristic curves of CsPbIBr_2_ PSC with and without moth-eye nanostructures at (**a**–**d**) different temperatures and (**e**–**h**) different angles of illumination.

**Figure 6 nanomaterials-11-02726-f006:**
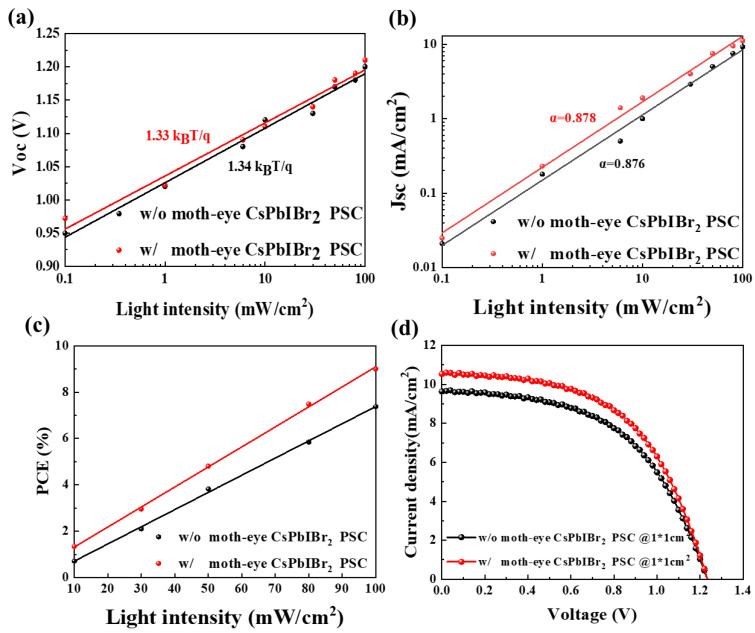
(**a**–**c**) Characteristic curve at different light intensity and (**d**) J–V curve of CsPbIBr_2_ PSC with and without moth-eye nanostructure.

**Table 1 nanomaterials-11-02726-t001:** Photovoltaic parameters of CsPbIBr_2_ PSCs of small and large active areas.

CsPbIBr_2_ PSCs with Moth-Eye Anti-Reflector	V_OC_ (V)	J_SC_ (mA/cm^2^)	FF (%)	PCE (%)	Improvement	
					J_SC_ (%)	PCE (%)
w/o (8.5 mm^2^)	1.27	10.89	65.83	9.17	9.37	9.92
w (8.5 mm^2^)	1.28	11.91	66.23	10.10		
w/o (1 cm^2^)	1.22	9.21	54.32	6.38	15.09	14.75
w (1 cm^2^)	1.22	10.60	54.16	7.01		

## Data Availability

The data is not available due to further study.

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
