# Peer review of "Performance Enhancement of All-Inorganic Carbon-Based CsPbIBr2 Perovskite Solar Cells Using a Moth-Eye Anti-Reflector"

_nanomaterials, 2021, doi:10.3390/nano11102726_

Round 1

Reviewer 1 Report

The authors reported the effectiveness of nano-imprinted bionic moth-eye structure acting as an anti-reflector, which is interesting.  This manuscript contains some interesting approach that grabs the reader's attention. Thus, this reviewer thinks this manuscript can be published in this journal after the minor modifications suggested below.

1. To characterized the effectiveness of the moth-eye reflector, the authors selected a device architecture containing carbon electrodes. How about using the normal NIP structures? 

2. Authors commented that all-solid CsPbIBr2 PSCs show long-term stability compared with organic-inorganic hybrid PSCs. However, there were no stability data in this manuscript. How about long-term stability? Do moth-eye reflectors help long-term stability?

Author Response

Dear Reviewer,

Thank you very much for your review dated 23th Sep. 2021, regarding the reviewer’s comments on our manuscript entitled “Performance enhancement of all-inorganic carbon-based CsPbIBr2 perovskite solar cells using a moth-eye anti-reflector”. We are grateful for the patient corrections from the reviewers, which have greatly improved the quality and significance of our manuscript. According to the comments of the referees, we have made appropriate revisions as present below. And the revisions are also highlighted in the revised manuscript. The detail response can be found in attach file.

Reviewer 2 Report

The author reported an efficient all-inorganic carbon based CsPbIBr2 perovskite solar cell (PSC) using a moth-eye anti-reflector. The author precisely described and discussed the importance of moth-eye anti-reflector and verified with FDTD simulation. This is very interesting and useful contribution for the possible application of wide-range wavelength anti-reflector in single and multi-junction tandem solar cells. I do believe, the present work may guide further development of CsPbIBr2 based PSCs. After carefully evaluation this manuscript, I suggest a minor revision before it is worth suitable for publication in the Nanomaterials. Therefore, before consideration of publication following issues should be addressed:

  1. The following related notable reports should be cited precisely.

    https://doi.org/10.1016/j.cej.2021.131831
    https://doi.org/10.1002/solr.202100509

  2. With the rapid improvement of PSCs, long-term operational stability has become a major concern for their commercialization and market adoption. However, the author did not provide any stability data. Light soaking, or moisture or thermal stability data should be provided, which will help the readers to understand this work more precisely.

Author Response

Dear Reviewer,

Thank you very much for your review dated 4th Oct. 2021, regarding the reviewer’s comments on our manuscript entitled “Performance enhancement of all-inorganic carbon-based CsPbIBr2 perovskite solar cells using a moth-eye anti-reflector”. We are grateful for the patient corrections from the reviewers, which have greatly improved the quality and significance of our manuscript. According to the comments of the referees, we have made appropriate revisions as present below. And the revisions are also highlighted in the revised manuscript. The detail response can be found in attach file.

Reviewer 3 Report

I have some comments regarding to submitted manuscript “Performance enhancement of all-inorganic carbon-based CsPbIBr2 perovskite solar cells using a moth-eye anti-reflector” by W. Lan and oths. : 

  1. The most acronyms in the text are not explained (or not properly explained): FDTD (lines 25, 31, 102, etc.), EQE (line 68, 149, 178, etc.), FF (194 199. Table , TPC and TPV (line 155), HTM (line 55). I’m sure the authors know, that the acronym should be explained as soon as it “is introduced” in the paper. Also, the short explication of the term “OrmoStamp“ ( Nanoimprint lithography process based on OrmoStamp (Micro Resist Technology GmbH, Germany) and corresponding materials would be useful. The chapter 1. “Introduction” could be supplemented by some references on the use of OrmoStamp technology (eg. : H. Park and oths, Microelectronic Engineering 98 (2012) 130–133; A. Amalathas, M. Alkaisi, Materials Science in Semiconductor Processing 57 (2017) 54–58; Graczyk and oths, Microelectronic Engineering 190 (2018) 73–78; R. Schmager and oths, Solar Energy Materials and Solar Cells 201 (2019) 110080).
  2. It  would be good to improve the quality of Figs. 1-6: the text entries in the Figs are blurred; In Fig.1f ticks on the right y-axis did not coincide with corresponding numbers.
  3. The sentence (lines 94-98): ”… Here, the resulted JSC of carbon-based CsPbIBr2 PSCs have been promoted by 9.4% from 10.89 mA/cm2 to 11.91 mA/cm2, which is the highest among PSCs with a typical structure of glass/FTO/c-TiO2/CsPbIBr2/Carbon, and the PCE of device was enhanced by 9.9% from 9.17% and 10.08%...” should be removed from “Introduction”: this one is repeated both in Abstract (lines 18-21) and Conclusions (lines 305-308).
  4. Purity grades of Absolute alcohol (AR, 299.7%) – line 110 and Acetone (AR, 299.0%) – line111 –should be corrected.
  5. In the sub-chapter 2.2, titled “ Preparation of CsPbIBr2 PSC and FDTD simulation”, there is no description of the FDTD method.
  6. The sub-chapter 2.3 “Fabrication of moth-eye on the device” should be revised: all the moth-eye layer formation steps should be presented consistently and clearly. The current description looks like a “copied” brief instruction that the manufacturer usually attaches to a process flow guide: line 133  - “Place the hydrophobically treated silicon template… and use an injection needle…., and put 20 drops on the silicon template”; line 142-  “Place the processed PSC on the glass substrate…, and use the same process steps ….”. Maybe, the more convenient way would be to present a graphic scheme of the steps of the moth-eye film formation procedures on the PCS device?
  7. Line 154 – the sentence “A scan rate of 50 mV/s was adopted” is duplicated: “A scan rate of 50 mV/s was adopted in the J–V measurement”.
  8. In the last part of the paper (lines 283 -298 and Fig.7) FDTD simulation results are presented: “…In addition, to further exploit the potential of moth-eye nanostructure, we established the optical model and performed the simulation optimization by FDTD tool.” In my opinion, the relevance of this part of the work in the context of the whole study is rather questionable. How these FDTD calculations relate to the objects under study? Usually FDTD simulation is applied for nano-optical device modeling (predicting) and/or for an analysis of light energy distribution in real system.  In the reviewed paper FDTD method was used for simulation of two theoretical  models (for the single and for the composite moth-eye nanostructures). The authors did not  specify whether the parameters of systems under study were used for FDTD  models elaboration. On the other hand, the authors also did not discuss any correlations between the elaborated models and the studied (measured) systems. I think that conclusions based only on the FDTD simulations should be removed from the Abstract. In my opinion, this part of the paper should be revised to be more in line with the research carried out  or it should be removed from the paper. Inter alia, there are no References on the FDTD method in the paper.
  9. Items from Nr.47 to Nr.55 in the list of References are not discussed (mentioned) in the paper text and should be removed.

 I would recommend to the authors to carefully verify and revise the study.

Author Response

(The authors gave the same response as above.)

Round 2

Reviewer 3 Report

The authors took into account the comments of the first review and made the necessary corrections.